:ۻ: PLOS | ONE

# Describing the indescribable: A qualitative study of dissociative experiences in psychosis

**Emma Černis** [1]*, **Daniel Freeman**[1], **Anke Ehlers**[2]

**1** Department of Psychiatry, Oxford Cognitive Approaches to Psychosis, University of Oxford, Oxford, United Kingdom, **2** Department of Experimental Psychology, Oxford Centre for Anxiety Disorders and Trauma, University of Oxford, Oxford, United Kingdom

\* emma.cernis@psych.ox.ac.uk

## Abstract

### Background

Despite its long history, dissociation remains under-recognised clinically, partly due to difficulties identifying dissociative symptoms. Qualitative research may support its recognition by providing a lived experience perspective. In non-affective psychosis, identification of dissociation may be particularly important given that such experiences have been implicated its development and maintenance. Therefore, this study aimed to understand in the context of psychosis: what it is like to experience dissociation; the impact dissociation might have; what factors begin, maintain or end dissociative experiences; and what beliefs people hold about dissociation.

### Methods

Qualitative interviews were carried out with twelve NHS patients with non-affective psychosis diagnoses and experience of dissociation. Data were analysed using Thematic Analysis.

### Results

Dissociation involves subjective strangeness, unreality, disconnection, and shifts in perception. It impacts on mental health (including psychotic experiences), daily functioning, emotional connection, and can lead to social withdrawal. Stress, fatigue, and excessive internal focus may be involved in development and maintenance. Participants found it very difficult to describe the experience of dissociation, and, as a result, often did not mention it to others. Even when shared, interviewees reported that their descriptions were misunderstood and therefore they did not receive information or support specific to dissociation. The consensus was that experiences of dissociation are negative, but that understanding them better helped to enable coping.

### Conclusions

The core subjective experience of dissociation appears to be a felt sense of anomaly (FSA), and we therefore suggest clinicians proactively enquire about such experiences.

**Citation:** Černis E, Freeman D, Ehlers A (2020) Describing the indescribable: A qualitative study of dissociative experiences in psychosis. PLoS ONE 15(2): e0229091. https://doi.org/10.1371/journal.pone.0229091

**Data Availability Statement:** Data cannot be shared publicly because of the terms and conditions contained within the ethics permissions granted for this study from the NHS Research Ethics Committee and Health Research, and

consented to by participants. Interviews were confidential to enable freedom of expression by participants, and participants consented into the study with the understanding that only anonymised quotations would be publicly available: not the entirety of the transcripts. Therefore, only illustrative excerpts from the transcripts, which qualify as the minimal data set, are included in the paper.

**Funding:** This study was funded by a Wellcome Trust Clinical Doctoral Fellowship awarded to EČ (102176/B/13/Z https://wellcome.ac.uk). AE is funded by the Wellcome Trust (200796 https://wellcome.ac.uk). DF is supported by an NIHR Research Professorship (https://www.nihr.ac.uk). The views expressed are those of the authors and not necessarily those of the NHS, the NIHR or the Department of Health. The funders had no role in study design, data collection and analysis, decision to publish, or preparation of the manuscript.

**Competing interests:** The authors have declared that no competing interests exist.

Dissociation is distressing, and has multiple impacts, but can easily be overlooked due to difficulties describing it and behavioural similarities to negative psychotic symptoms such as withdrawal.

## Introduction

Dissociation is a complex phenomenon in mental health, with a large body of historical and contemporary publications outlining many (often opposing) frameworks [1]. Despite this wealth of literature, dissociative symptoms remain repeatedly under-recognized or misidentified [2]. Recognition of symptoms may be a particular challenge in dissociation due to the ongoing conceptual debate, which has negatively impacted the clarity of the construct [1,3], and because many clinicians receive inadequate training in this area [4].

Nevertheless, dissociative experiences have been demonstrated to occur across a range of psychiatric disorders [5], and may therefore be considered trans-diagnostic at least to some extent. Indeed, dissociative experiences have been found at an 'elevated level'–up to fifty percent—in psychosis [6]. This is of interest given ongoing discussions about the extent of symptom overlap between psychosis and dissociation [6], and particularly given the suggestion that anomalous experiences–such as dissociation–are important in the formation and maintenance of psychotic experiences such as hallucinations [7] and persecutory delusions [8]. For example, the Garety [9] model of positive psychotic symptoms suggests that anomalous experiences drive a 'search for meaning', resulting in the adoption of beliefs that may form the basis of delusions.

Improvement of the recognition of dissociation is therefore an important goal clinically [2] as well as in psychosis research, where it would be potentially beneficial to investigate dissociation as a putative causal or maintenance mechanism. However, difficulties identifying dissociative phenomena impede these efforts. Qualitative research may go some way to improving this situation: such methods 'answer questions about experience, meaning and perspective' [10], and therefore give an opportunity for patient voice to be heard by clinicians and researchers. Surprisingly, there is little qualitative evidence regarding the lived experience of dissociation to guide our understanding, and none regarding the experience of dissociation in the context of psychosis. The latter would be of particular interest and utility given the findings of a significant overlap between psychotic and dissociative symptoms [6]. Therefore, this study sought to further our understanding of dissociation by referring to patient expertise–and specifically patients with lived experience of both dissociation and psychosis.

The current study aimed to answer the following questions:

1. What is it like to experience dissociation (in the context of non-affective psychosis)?

2. What is the impact of dissociation?

3. What factors begin, maintain, and end dissociation in the context of non-affective psychosis?

4. What thoughts and beliefs do people with non-affective psychosis have about their dissociative experiences?

These questions were chosen to learn more about the phenomenology of dissociation in psychosis (question 1), and illustrate what it is like to live with (questions 2 and 4). To facilitate clinicians' understanding of the problem, questions 3 and 4 aim to explore the experience of dissociation in psychosis using a cognitive-behavioural framework, which is the most familiar psychological approach for many UK clinicians.

## Methods

This study received ethical approval from the NHS Health Research Authority (REC reference: 18/SC/0048). All patients gave written informed consent to participate, including for the interviews to be audio-recorded and anonymised quotations to be used. Capacity to consent was established by the first author, a qualified clinical psychologist.

### Participants

Twelve participants were recruited from Oxford Health NHS Foundation Trust via referral by clinical or research staff. Staff referred patients on their caseload who may have already reported dissociative experiences, or may be interested in participating. After confirming with the first author whether the person provisionally met entry criteria, staff approached the person with a participant information sheet to discuss the study. Those who were interested and felt the study may be relevant to them gave verbal consent for the first author to contact them directly to discuss the study in further detail and arrange to meet to check eligibility to participate according to the inclusion and exclusion criteria. A summary of participant recruitment is shown in Fig 1.

Inclusion criteria were: a primary diagnosis of a psychosis spectrum disorder (confirmed by the clinical team at referral, and recorded according to self-report); aged 16 or over; self-reported experience of dissociation; and a Dissociative Experiences Scale (DES-II; [11]) score above 5.4. The cut-off score on the DES-II was chosen based on Carlson and Putnam's statement that a score of 5.4 is a typical mean for the general population. This cut-off was chosen because we were interested in exploring a wide range of perspectives: including those who had since improved and were no longer regularly experiencing dissociation. When completing the DES-II, participants were encouraged to consider more recent experiences (i.e. the past two weeks) in order to elicit dissociative experiences which would be easy to recall and discuss. Participants with general population-level scores (i.e. lower levels of recent dissociation) were able to recall previous dissociative experiences clearly and discuss them in detail. In these cases, the DES-II score reflects that the participants had not experienced dissociation within the past two weeks, not that they had never experienced higher levels. It is of note that the majority of this sample (nine participants) scored far above the proposed average mean for Schizophrenia (DES-II = 15.4) and within the ranges expected for dissociative disorders [11]. The mean score was 36.91 (SD = 20.00), indicating high levels of currently-experienced dissociation within the group. 'Self-reported experience of dissociation' was assessed via discussion between participant and the first author, using the items of the DES-II and the content of the Participant Information Sheet (PIS) as prompts. The PIS featured a selection of eight items from the Dissociative Experiences Measure, Oxford (DEMO; [12]) and also stated: '*Sometimes, people have strange feelings and experiences–like feeling that they are 'spaced out', numb, unreal, disconnected, or 'trapped in a bubble'. These experiences are sometimes called 'dissociation' or 'depersonalisation'*'. The term 'depersonalisation' was included in this instance since recent popular articles [e.g. 13] have used this term and it may therefore have been heard by participants before. In the discussion regarding self-reported experience, participants were asked if they identified with any of the content in the PIS or DES-II, or had experienced anything similar that was not listed. They were then asked to give some brief detail in order to determine that the item had been understood as intended.

Exclusion criteria were: insufficient comprehension of English to participate; primary diagnosis of alcohol or substance dependence; primary diagnosis of personality disorder; presence of organic syndrome or learning disability; or dissociation only experienced in the context of alcohol or illicit substance use. Additionally, two people were not invited to participate

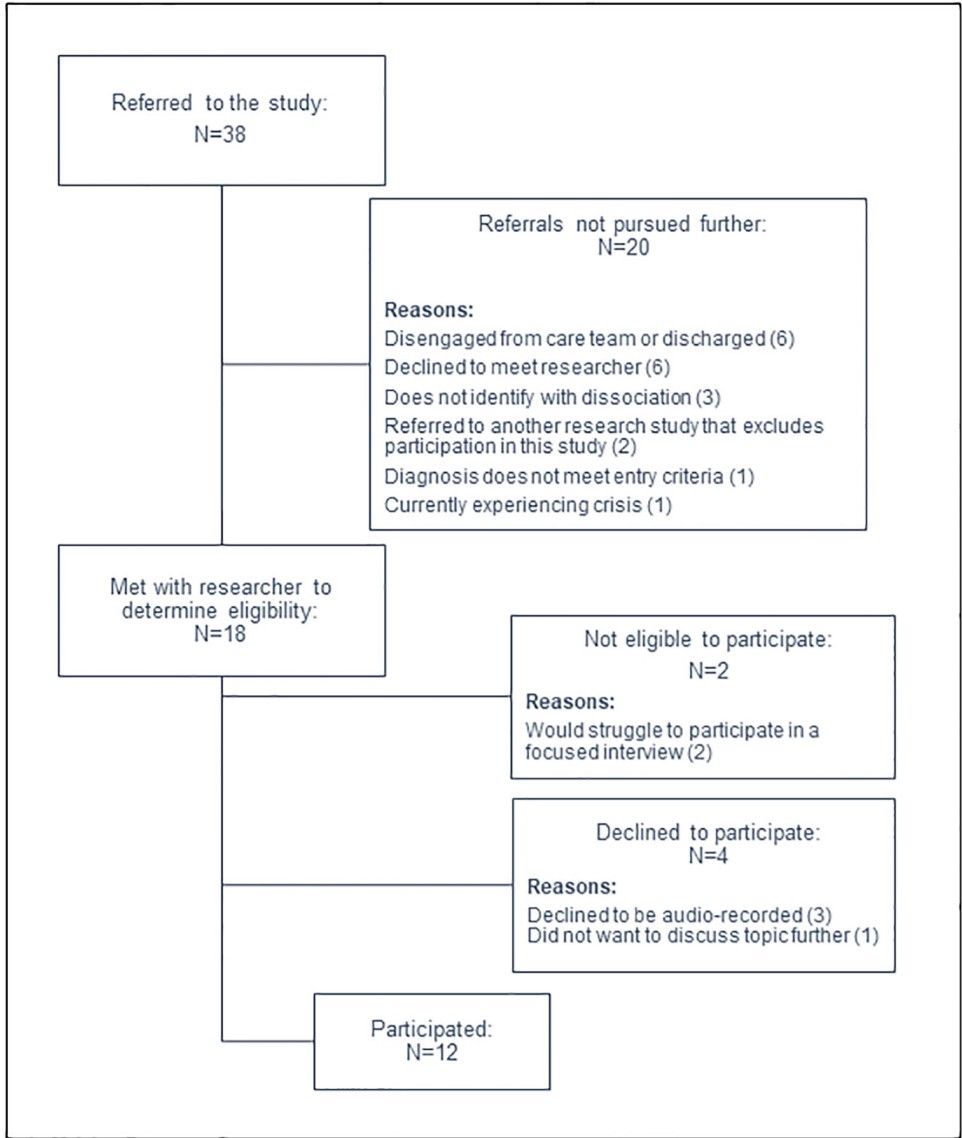

**Fig 1. Summarising the referral and recruitment details of the study.**

following clinical judgement of their ability to participate in a focused interview. At the initial eligibility meeting, one person was unable to identify or describe their thoughts or feelings about dissociation; and a second person struggled to disengage from other distressing topics in order to answer focused questions. Participant characteristics can be seen in Table 1 (below).

Participants were not directly asked about trauma history, nor was this screened for upon entry to the study. However, four participants disclosed life experiences or re-experiencing symptoms indicative of previous trauma during interview.

## Procedure

Interviews were semi-structured, audio-recorded, and conducted face-to-face by the first author. The first author is a Clinical Psychologist with experience working with psychosis,

**Table 1. Summarising participant characteristics.**

| Participant code | Gender; Age at interview | Ethnicity | Marital status | Employment status | Recorded diagnosis | DES-II score |
|---|---|---|---|---|---|---|
| 1 | M; 16 | White British | Single | Student | Psychosis NOS (First Episode) | 47.14 |
| 2 | M; 55 | White British | Single | Voluntary work | Schizoaffective Disorder | 7.14 |
| 3 | M; 29 | White British | Married | Self-Employed | Schizophrenia | 50.71 |
| 4 | F; 27 | White British | Married | Housewife | Schizophrenia | 9.29 |
| 5 | M; 40 | White British | Single | Unemployed | Schizophrenia | 34.29 |
| 6 | M; 64 | White British | Single | Retired | Schizophrenia | 42.86 |
| 7 | M; 18 | British Asian | Single | Student | Psychosis NOS (First Episode) | 33.21 |
| 8 | F; 33 | White British | Single | Unemployed | Schizoaffective Disorder | 52.86 |
| 9 | M; 42 | White British | Single | Self-Employed | Paranoid Schizophrenia | 38.93 |
| 10 | M; 47 | White British | Single | Unemployed | Paranoid Schizophrenia | 51.43 |
| 11 | F; 21 | White British | Single | Student | Unspecified Inorganic Psychosis | 68.57 |
| 12 | F; 43 | Mixed: Irish & African | Single | Unemployed | Schizophrenia | 6.43 |

trauma, and dissociation. Interviews took place at participants' homes or local NHS mental health team base, according to participant preference. Interviews followed a topic guide (see S1 File) which was developed via consensus meetings, consultation with a Lived Experience Advisory Panel (LEAP), and a pilot interview. The topic guide was drafted by the first author and discussed amongst all three authors during the planning stages of the study. The study proposal and materials were then taken to the LEAP. This panel was formed of people with experience of psychosis, and mental health services and treatments. It was initially formed by the McPin Foundation to advise on a clinical trial of a psychological intervention for psychosis. Members of this panel gave feedback on the information sheet and study design, including reviewing the topic guide. No changes were made at this stage, and the LEAP agreed with the importance of the research questions posed. One volunteer from the panel then took part in a pilot interview following the topic guide. The experience and content of this interview was discussed, and feedback subsequently incorporated into the approach taken during the study interviews (e.g., that prompting for further detail was acceptable to the interviewee); no content in the topic guide was changed. The topic guide was also reviewed by the first author following the first three interviews to ensure it was still suitable. The only change made at this stage was to use participants' DES-II forms as an initial point of discussion if they found it difficult to answer the first open-ended question about their experience of dissociation (below).

As can be seen in the S1 File, the topic guide was structured according to the four research questions outlined above (*Introduction*). Interviews therefore began with an open question asking the interviewee to describe their experience of dissociation using their own words; followed by questions relating to each of the four research questions listed above; and ended with an invitation to discuss any aspects of dissociation not already covered. Interviews lasted on average one hour (range: 30 minutes to 1 hour 40). The audio recordings were transcribed and anonymised, with transcriptions checked for accuracy against the audio recording.

## Method of analysis

NVivo version 12 [14] was used during the coding and analysis of data. Thematic analysis was the chosen analytic method since it has been identified as an approach well-suited to underresearched areas, enabling flexibility and detailed insight–including highlighting contradictions within the data [15]. Interviews were analysed using deductive thematic analysis to separate the data into the research questions posed, and then using inductive thematic analysis

within each question to enable flexibility and fidelity to the data. A coding framework was developed by the lead author following immersion in the transcripts of the early interviews. The framework evolved through constant comparison and modification as each new interview was added to the dataset. One example of this is that initial interviews were coded with many fine-grained ('splitter') codes [16] (e.g. 'affects concentration', 'confusing', 'difficult to remember the truth'), which then began to be incorporated into fewer broader ('lumper') codes [16] (e.g. 'I can't trust my head') as similar content continued to appear in later interviews. Details of the original fine-grained codes were retained in NVivo to facilitate understanding of these broader codes. Broader codes were later gathered into 'themes' according to the thematic analysis method [15].

Interviews were analysed using a single primary coder, supported by discussions within the wider team about the methods involved (i.e. transitioning from fine-grained to broader codes). To reduce bias, time was taken between preliminary and review coding of transcripts [17], and preliminary themes were reported to the LEAP partway through the study. Additionally, negative case analysis was also undertaken: 'actively sought' and 'spontaneously occurring' [18] data that contradicted a code was incorporated or added as a new code in order to reflect the complexity of dissociative experience. For example, the sub-theme 'negative experience: not wholly negative' (below) arose from evidence contradicting the sub-theme 'negative experience: fear'. However, it is important to note using a single primary coder is a source of bias (see Discussion). The primary coder was the first author (EČ); a brief reflexive statement regarding the biases this author may have brought to the analysis can be found in the S2 File.

## Results

### 1. What is it like to experience dissociation?

Please see Fig 2 for a summary of the results for research question one.

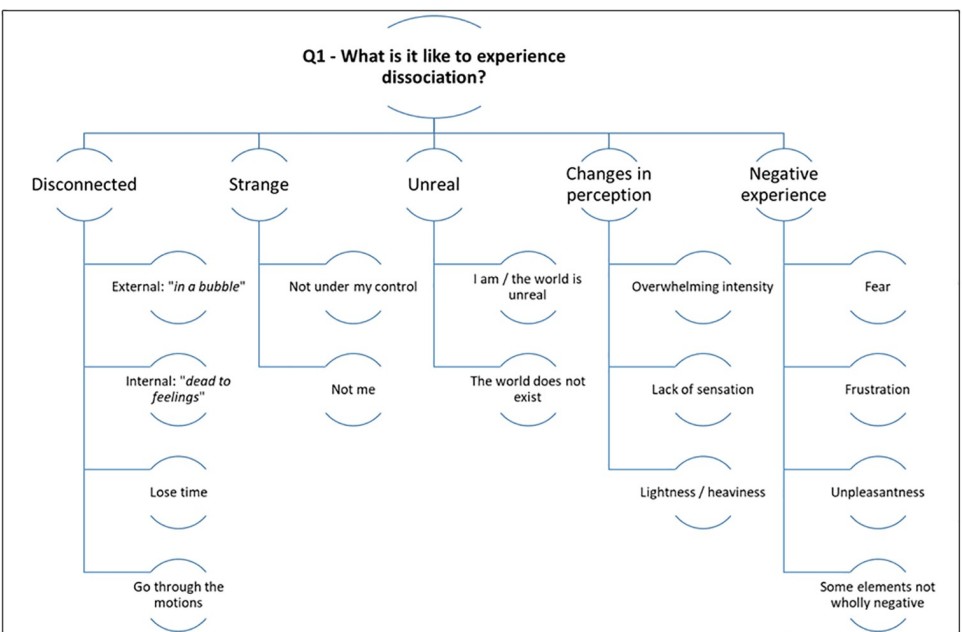

**Fig 2. Summarising the themes for the first research question.**

**Disconnected.** All participants described being "detached" and "disconnected". For some, this felt like being trapped in a metaphorical "*box*" (P8), "*bubble*" (P4), behind "*a layer of Perspex*" (P8) or experiencing life through a "*keyhole*" (P1). Others experienced this in a less definite way and described feeling "*murky*" (P6) or hazy. Whilst these sensations often related to external events, interviewees also reported detachment from internal cues: feeling "numb", having a "blank" mind, and being "*dead to feelings*" (P2).

Detachment resulted in losing time (P8: "*you just don't really feel like you're here . . . time kind of elapses*"), and feeling as though they were "going through the motions" of living whilst feeling none of the usual associated sensations:

P8: "*Sometimes I can even show the right emotions, I guess, but I don't feel them*"

Feelings of disconnection also resulted in sensations of being separate from, or outside of, their body:

P5: "*. . .and suddenly I feel a bit outside of myself, dizzy, and kind of disoriented*"

Participants reported that these experiences of disconnection persisted, sometimes building in intensity, and often also felt strange and unreal (below).

**Strange.** Ten interviewees described the quality of the experience of dissociation as "strange", "weird" or otherwise "*a bit disconcerting*" (P3). Two related elements appeared to contribute towards this. The first is the issue of choice or control. Participants reported not having a choice as to whether or when dissociation would occur; not having control over it once it began; and sometimes not feeling fully in control of their actions when feeling dissociated:

P5: "*You try and get control of it, but it's like a bar of soap or something, it just keeps slipping out of your hands.*"

Secondly, people described dissociation as though it was a separate entity to themselves, including not identifying themself as "them" at times:

P7: "*It doesn't feel like me, but deep down I do know it's me. It's not a case of: I think I'm watching through someone else–it just doesn't feel like me.*"

P6: "*Like now, I'm watching myself speaking, but that's not me.*"

Interviewer: "*Which bit isn't you?*"

P6: "*The speaking bit. The words that are coming out seem to be coming from me, but there is something here sort of monitoring the words and saying, 'well I suppose he's talking about the right sort of thing'*

**Unreal.** Sensations of strangeness and disconnection appear to contribute to another major theme of 'unreality'. Eight people described that their surroundings, experiences, or they themselves did not feel real, or that they struggled with reality:

P6: "*It's as if what appears to be the reality around me is crumbling*"

Four people even stated that they felt as though the world around them might not exist at all.

Participants described being trapped in a "*simulation*" (P1), computer (P10), video game (P9, P12), nightmare (P2), or alternate universe (P6). These examples of artificial–unreal–environments encapsulate sensations of strange unreality (externally) and the apparent lack of personal agency (internally).

Perhaps unsurprisingly, given the above, some participants described experiencing life as fragmented:

P1: *"Life isn't a coherent succession of events any more. It's just different sort of sporadic moments which I just remember"*

This was a sensation participants doubted anyone else had felt before:

P6: *"I can't imagine anyone else having the same sort of consciousness"*

**Changes in perception.**   Interviewees commonly reported increased volume and intensely bright colours as part of their dissociative experience. Numerous reports blur the lines between anomalous perceptual experience and hallucinations:

P3: *"It's sort of like old 3D films: it's got extra colours around the images."*

Understandably, some interviewees found these experiences overwhelming. This "*over-whelm*" (P2) appeared to drive the "unreality" discussed above, and linked closely with the reports of being unable to cope, "*hypersensitive*" (P9), or "*fragile*" (P2).

In terms of physical perception, participants reported a lack of bodily sensation, feeling separate from the body, or even feeling they did not have a body at all:

P12: *"And it didn't feel like it was my body, it just felt like- I was just sort of reflecting. Like I was a mirror or a camera or something like that. I wasn't aware that I actually had a body."*

Interviewer: *"But then you sit and relax and then your body starts to feel unreal*?

P10: *Yeah, like it's not there.*

Interviewer: *[. . .] So it's like a whole leg has gone missing or..?*

P10: *Sometimes, yeah. My arm or something, yeah."*

This lack of sensation (or absence of a body part) linked with sensations of lightness or floating:

P6: *"It feels- I wouldn't say weightless, but you don't- there's no sensation"*

Conversely, some people described dissociation in terms of heaviness, slowness or tiredness:

P3: *"It can feel like trying to move through water fast"*

Some described the sensation in terms of depth or distance (which at times overlapped with feelings of numbness):

P6: *"I don't feel I'm outside my body, I feel I'm deep inside"*

Anomalous bodily and perceptual experiences further contributed towards difficulties feeling connected to one's self:

P9: *"I'm present in this body in emotions and stuff, but it ain't me: just a thing. [. . .] Because I don't use my facial muscles for expressions, when I do, it's like 'oh that's someone else, no, I'm not comfortable with that'."*

P10: *"And when I look in a mirror as well, I look like someone else all the time. I see a different person in different mirrors."*

**Negative experience.**  The vast majority of interviewees felt that dissociation was a negative experience:

P8: *"it was the worst feeling in the world".*

Specifically, participants reported fear and frustration:

P2: *"Everything was frightening. And I was fearful and anxious and paranoid."*

P4: *"Always like a frustration. I'd get so frustrated that it was happening again."*

Participants also predicted that other people would be scared, upset, or worried if they understood the truth about dissociation:

P6: *"I would like my head to be opened up and for me to say, 'look, this is what it's like inside', and I think people would be very frightened."*

However, whilst many described the unpleasantness of dissociation, some interviewees explained sometimes the experiences could also be exciting or refreshing:

P8: *"Honestly? Sometimes [it's] quite refreshing because- I think the nature of how my illness plays out sometimes that I over-care, so for a while it's quite refreshing. [. . .] It's quite like, 'oh okay, this is how it feels when you don't really give a shit' [. . .] And at first it's like, 'okay, it's quite cool, I get it', but then it's not cool. It's not cool."*

Many interviewees also described how they had become accustomed to some of their dissociative experiences, expressing apparently neutral emotional experience of dissociation:

P7: *"By the evening, I'll still be a bit murky–but I'm usually murky, so it's fine."*

However, Participant 6 gave an insight into how it may not be entirely neutral:

P6: *"I'm so used to it I can talk about it quite blandly, although it is quite an intense feeling. I'm overwhelmed by it all the time. But because I'm so used to it, it doesn't bother me quite the same way it used to."*

## 2. What is the impact of dissociation?

Please see Fig 3 for a summary of the results for research question two.

**Functioning.**  Eleven participants reported some minor difficulty with day-to-day functioning. Participant 1's summary *"I mean–I can function, it's just hard"*, appeared to be the

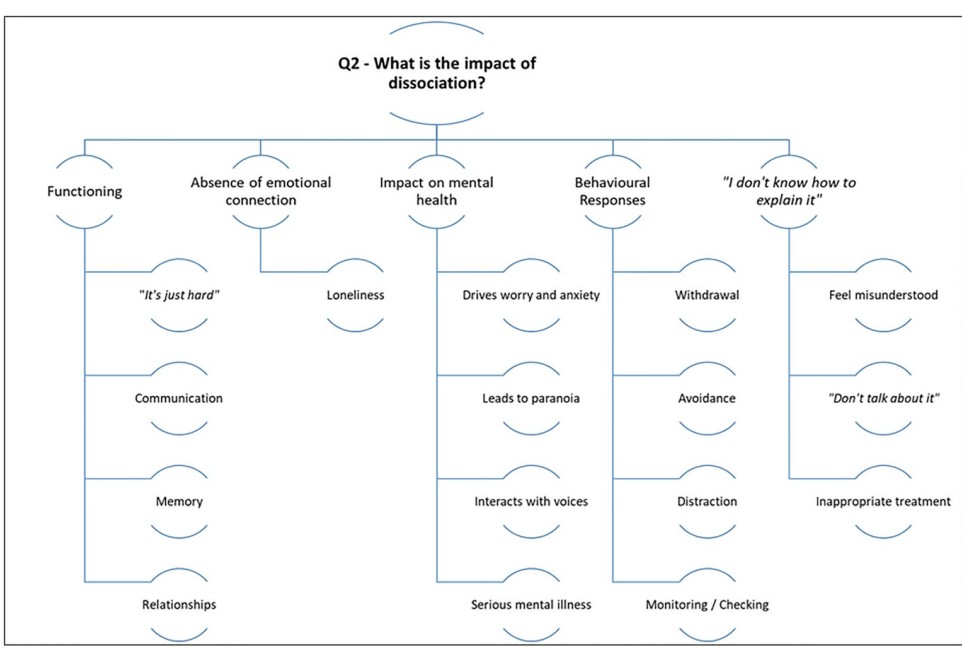

**Fig 3. Summarising the themes for the second research question.**

general view. However, three interviewees reported dissociation had significantly disrupted their life (P4: *"I didn't finish my A Levels"*; P6: *"I had a year off school"*).

Interviewees described a number of specific 'in-the-moment' impairments which were associated with greater frustration or distress, such as communication (comprehension or speech generation). Two key ways in which memory was affected by dissociation were described: firstly, participants experienced absence of memory or difficulty retrieving memories:

> P1: *"If I go into a room, I forget how I got there or I just forget large chunks of what I do throughout the day [. . .] I feel like it's there, it's just being blocked."*

Secondly, interviewees described feeling a lack of ownership or agency in retrieved memories:

> P2: *"With some things, it's like I have the memory of something, but it's more like someone's described it to me in detail, rather than me experiencing it."*

Memory and communication difficulties also impacted social relationships. Participant 7 described feeling concerned that if friends discovered he could not recall meeting with them, the friends would be upset and conclude that he did not value their friendship. Concerns about inadvertently offending people in this way–or because of being under-responsive or non-communicative when feeling dissociated–were common. Participants expressed not wanting to upset or stress their friends and family, as well worrying what people who knew them less well would make of their dissociated state. However, close friends and family can begin to recognise the signs that the person is experiencing dissociation:

> P8: *"She says she can always tell because I just- she says, 'You're just not there, it's blank, like– there's no character, there's no spark, there's no nothing. It's like your brain's just switched off from us'."*

**Absence of emotional connection.**    Participants reported that if they did notice dissociation happening (many reported that they did not notice until it ended), they tried to *"snap out of"* (P11), *"break through"* (P8) or *"get out"* (P3) of it. Often, this was to try and re-connect with others: one of the most widely endorsed themes regarding the impact of dissociation was 'loneliness'. Seven people reported feeling *'just really lonely'* (P4) and isolated. Although this was explained as being because '*no-one understands*' (P10) by a number of participants, most stated that it was a result of being unable to feel emotional closeness because of dissociation:

P4: *"I would just burst into tears or just feel- and you know you're not alone, because you've got everybody around you [. . .] but at the same time, in that moment, and in that bubble, you just feel you are the most disconnected, isolated person ever."*

**Impact on mental health.**    For some, the desire to "snap out of" dissociation was also because of the impact it can have on mental health:

P11: *"[Dissociation] can get quite linked to my mood or to a psychosis episode, so it's important that I try and break out of it as early as I can."*

Most interviewees made a link between dissociation and anxiety, indicating that dissociative experiences lead them to worry more:

P12: *"Being cut off from the real world and just sort of chattering away on my own, or staying in at home and over-thinking and ruminating and not actually–yeah, feeling part of the world."*

There were also indications that this could tip into paranoia:

P8: *"I think the paranoia is where the overlap is [. . .] it's very easy for the voices and the paranoia to creep in and fill your head with what could have happened, rather than knowing what did happen."*

Participant 8 was not the only person who discussed the connection between voices and dissociation. Others reported that dissociative episodes could act as a trigger for the voices starting (P11), or make them louder and more aggressive (P4). Participants 2 and 6 described hearing non-voice sounds which became part of the perceptual *"overwhelm"* described above, and which they interpreted as having a particular meaning or significance:

P6: *"My hearing can change; I can hear things and I think 'What does that mean?' As if somebody's trying to communicate with me a bit sometimes."*

Dissociation can therefore exacerbate worry and anxiety, which may interact with psychotic symptoms. A number of interviewees described a next stage in the sequence of events: yet more severe consequences for their mental health:

P2: *"Well, I became suicidal actually"*

**Behavioural responses.**    Participants described a range of methods for coping with dissociation. The most commonly reported was the impulse to withdraw:

P4: *"It's really hard because some days, you do just want to hide."*

When withdrawal is reported as being under the person's control–rather than "zoning out" and *"just sit[ting] and star[ing] at nothing"* (P4)–it can take the form of having a *"lie down"* (P6), or finding another comfortable way of *"rid[ing] it out"* (P1) or *"let[ting] it go away"* (P7). This kind of withdrawal was linked to the belief that there is no way of preventing or stopping a dissociative experience and therefore the only option is to wait for it to pass.

Withdrawal was also related to the belief that stress and other intense emotion are negative and best avoided (see Question 3). This resulted in avoidance of busy social situations, sticking to familiar routines, and trying to stay "in the moment" to avoid becoming stressed:

> P6: *"If I can just smoothly move through the experience and not allow this hook- [gesticulates near head] -not be too aware of what's going on, just allow it to flow through, then I can manage."*

Sometimes, therapeutic coping mechanisms–particularly grounding and sensory kits–were employed to achieve avoidance of stress. However, participants noted that while these sometimes reduced the length of time spent dissociated, they did not prevent it happening entirely. Others noted that they forgot coping techniques when dissociated:

> P12: *"So one minute it's like "oh..!", next minute it's like. . . [makes "whoosh" motion]. It happens so quickly [. . .] and you forget that: "oh, I could just phone my CPN [Community Psychiatric Nurse]". I mean, you just forget things like that."*

As well as physical avoidance of potentially stressful situations through withdrawal, participants also described mental avoidance of difficult thoughts or thoughts about dissociation through distraction and "keeping busy". Techniques employed included fostering a busy lifestyle, asking close friends or family provide distraction through conversation, or performing mental tasks or puzzles:

> P7: *"Plan meals for either the week or the next two weeks, even if I have no intention of cooking them"*

Mental tasks sometimes took the form of devising tests to see whether dissociation had passed or to try and force it to pass. For example, Participant 8 recounted how they forced themselves to look at their friend's pet snake to try and invoke a fear response and "break" dissociation.

Mental avoidance of dissociation was also achieved by not talking about the experience (P3: *"I tend not to speak about it much"*). As well as feeling that no-one would understand the experience (next section), this was driven by fear of dissociation:

> P6: *"I think it was really I was just terrified of what was going on and I didn't trust anyone enough to talk to them about it"*

Despite this high level of mental avoidance, participants also implied that they actively observed or judged their internal world during dissociation, sometimes engaging in monitoring or checking behaviours:

> P7: *"A case of "ok, remembering this is not real". Well, I always knew it wasn't real, but it's a case of: I can see it happening, I can just sort of move past it–it's not as if I have to stop and look at it, which was the issue before, because I needed to make sure whether or not it was happening."*

**Not knowing how to explain the experience.**    The difficulty of describing dissociative experiences and finding the appropriate words or language for them was a common theme of the interviews. Many relied upon metaphor to get their meaning across (such as the "box", "bubble" etc. above). Ten of the twelve participants explicitly stated:

P5: *"I don't really know how to explain it"*

It was therefore also a common experience for interviewees that other people did not understand what they were describing:

P8: *"When you explain it, nobody gets it"*

This led to people choosing not to speak about dissociation, and–if they did–professionals mistaking it for another problem, particularly depression or psychosis:

P6: *"They might be related, but they're very separate experiences to me. I think that's fundamental. When you're actually going through the problem, it's very difficult to get across to somebody that you're having two separate qualitative experiences. I've got the mysticism [dissociation], and I've got the schizophrenia."*

As well as feeling misunderstood and unheard, participants suggested that people experiencing dissociation may undergo inappropriate or irrelevant treatment (such as Participant 12, who described being on medication for bipolar disorder: *"obviously I didn't have the symptoms, so it was working on just 'me'."*).

## 3. What factors begin, maintain, or end a dissociative experience?

Please see Fig 4 for a summary of the results for research question three.

**Factors that begin dissociation.**    Whilst some participants stated that dissociation *"can happen completely randomly"* (P1), the majority of participants noticed that anxiety or "stress" could trigger a dissociative experience. This was true for emotional distress more broadly; including low mood, feeling "upset", or worsening psychotic symptoms. For those who experienced voice-hearing, a number had heard voices (or "noises") long before they had felt disconnected, one person suggested that the two came together (although they believed them to be separate phenomena), and others indicated that feeling disconnected allowed "space" for voices to intrude and worsen. A similar pattern was reported for paranoid thinking: some participants cited paranoia as causing stress and anxiety, which in turn caused them to dissociate; whilst others described feeling paranoid as a result of feeling unreal or feeling that something had been altered in themselves or their surroundings.

Participants reported that listening intently to voices, *"dwelling on a situation"* (P4) or *"getting into your own head"* (P5) by focusing on thoughts and emotions could cause dissociation. This included dwelling on dissociation itself, in which case dissociation was seen to *"trigger itself off"* (P4).

Related to internal focus, participants reported that dissociation was more likely to happen when they were alone or feeling isolated. Isolation was also noted as something that may have initially contributed to dissociation alongside stress: nine participants described dissociation beginning during a significantly stressful period in their lives–and three made explicit reference to feeling unsupported or alone during these times. These stressful periods included surviving a domestically violent relationship, having a *"turbulent"* family life when young, being

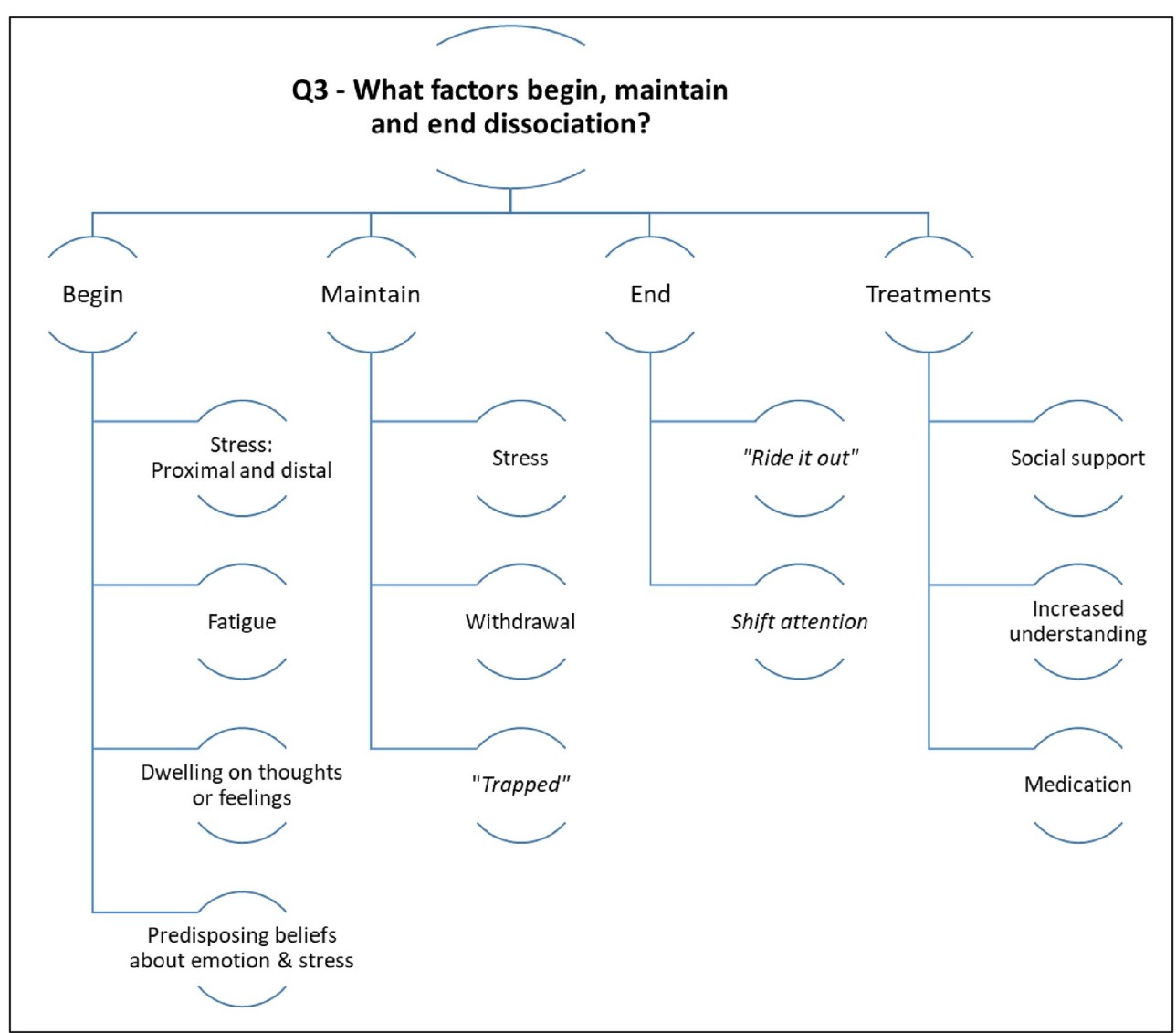

**Fig 4. Summarising the themes for the third research question.**

physically attacked, experiencing bereavement, and having to drop out of university due to mental ill-health and stress.

Proximal precipitating factors identified included feeling "*mentally exhausted*" (P1), "*sitting doing nothing in particular*" (P6), and feeling "overwhelmed" by too much mental or perceptual stimulation (including from psychotic experiences). "Feeling overwhelmed" was discussed with particular reference to emotion, which six participants reported that they "*didn't want*", since "*things like emotions I've never been able to cope with*" (P9). Such discussions also highlighted longstanding beliefs that intense emotion was potentially dangerous due to participants' relative "fragility" and (in)ability to cope.

These participants explained how they tended to avoid or "*repress every single emotion*" (P11), sometimes due to beliefs about the danger of emotion:

P9: *"Yeah, no, I didn't want emotions. From the earliest I can remember."*

Interviewer: *"Any particular reason why you didn't want emotions?"*

P9: *"I'm uncomfortable with them."*

Interviewer: *[. . .] "What would happen if you didn't stop [your emotions]?"*

P9: *"Well, they can become so intense that you have no control over it. [. . .] They'd be psychotic."*

This seems particularly relevant to the development of dissociation: some participants identified that emotion could bring on dissociation (P6: *"you get charged emotionally [. . .] and that can bring on layers of unreality as well"*) and therefore dissociation is *"a withdrawal from excessive emotion"* (P11). However, the exact chain of events is unclear, as some interviewees noted that avoiding emotion may actually have led to them experiencing dissociation (P11: *"it was some form of emotional leakage"*).

Other predisposing factors identified by participants were: a tendency to daydream excessively when young, having a busy or enquiring mind (prone to dissecting events), and poor sleep. Further, participants expressed negative self-beliefs of being "bad" or "different"; these may also be predisposing factors.

**Factors that maintain dissociation.** The factors of stress, tiredness, isolation, and dwelling on internal events identified above were also cited as relevant to the persistence and re-occurrence of dissociative experiences.

Additionally, withdrawal behaviour may perpetuate dissociation: participants reported disengaging from activities which might be potentially positive because they believed it would be impossible to feel the positive emotion. This arguably continues the absence of emotional connection:

P6: *"The few relationships I've had with women have been with women I've not really fancied particularly, because there's not a large emotion in my mind. If I feel a large charge of emotion, I have to back off."*

The way in which participants spoke about dissociation also implied two further potential maintenance factors, although these were less clear. The first is *"being trapped in"* (P12) the experience (including losing insight). Participants reported that as a result of "being trapped", they found it difficult to respond to dissociation, which allowed the episode to continue unchecked:

P11: *"There are dissociation images which are engaging ones, where I think I'm in them."*

This may interact with beliefs about control over dissociation. Secondly, previous positive experiences of dissociation might promote ambivalence about dissociation and affect how someone responds when they feel it coming on:

P6: *"In a way, it's sort of an adventure"*

**Factors that end dissociation.** Interviewees reported that dissociation can end suddenly or by slowly fading away. Two participants suggested that dissociation might never end, but rather, one's attention shifts elsewhere. Indeed, being distracted–particularly by other people talking directly to you–appeared to be one of the most common routes out of dissociation. When this was sudden, participants talked about being "snapped out of" dissociation.

Four participants reported that there was nothing that could be done to end dissociation (recalling the issue of control discussed above), and that all that could be done was to *"ride it out, sort of wait for it to finish"* (P1). However, two participants identified that they noticed dissociation had ended when they were *"not doing anything–just sort of trying to relax"* (P3).

In terms of treatment approaches, it was evident that participants welcomed support for dissociative experiences. Two participants suggested that a lack of medical intervention earlier in their lives had led to worse outcomes for their mental health, and seven participants noted that dissociation was easier to manage once they had achieved a better understanding and acceptance of it. The support of others–discussions with friends and family, or formal support groups–were cited as important for achieving this and for managing mental health more broadly.

Interviewees also named specific treatment techniques as effective including medication (P4: *"I take my medication religiously every day"*), using a box of sensory objects for grounding (P8: *"It reminds me of my Nan's dog's ears, which always seems to bring me back"*), and making lifestyle changes:

> P12: *"I started swimming regularly [. . .] There were things I was trying to do to change, like CBT or giving up smoking, trying to have less caffeine. . ."*

However, although these were often reported as being helpful, it should be noted that participants also found limitations to these treatment approaches. For example, some noted that grounding techniques might only reduce the length of dissociation, rather than preventing it; care plan or psychological techniques might be forgotten or inaccessible during dissociative episodes; and medication was not always effective for the specific experience of dissociation.

## 4. What thoughts and beliefs do people have about dissociative experiences?

Please see Fig 5 for a summary of the results for research question four.

**During the experience.** Some interviewees said that they are aware of their thinking changing during dissociation. Participant 6 described their thoughts *"going skewiff"* or *"wonky"*, and Participant 11 said that their thoughts might speed up or *"completely slow down"*. Five participants referred to "going blank" and having *"just nothing in your head"* (P8). These experiences made it challenging for people to identify any thoughts arising during dissociation.

When people could identify thoughts, these often reflected the sensations and experiences described in Question 1 (for example, feeling overwhelmed: P6: *"it's too much"*; or finding things unreal: P1: *"it's just like a game"*). These could occur alongside awareness of or insight into the experience (P9: *"Ah, you're going through this again"*).

Interviewees described focusing on what was happening, and anxiously searching for explanations:

> P10 *"Why me? What have these things happened to me?"*

Participants described either actively trying to figure out what was happening or passively observing the experience to learn more about it. Potential answers to these questions ranged from benign (P7 *"Am I dreaming?"*), to catastrophic (P4 *"I'm completely losing my mind."*). Some also appeared more psychotic in nature:

> P12: *"I'm not part of the world any more. I'm not going to get away to the other planet and live like everyone else."*

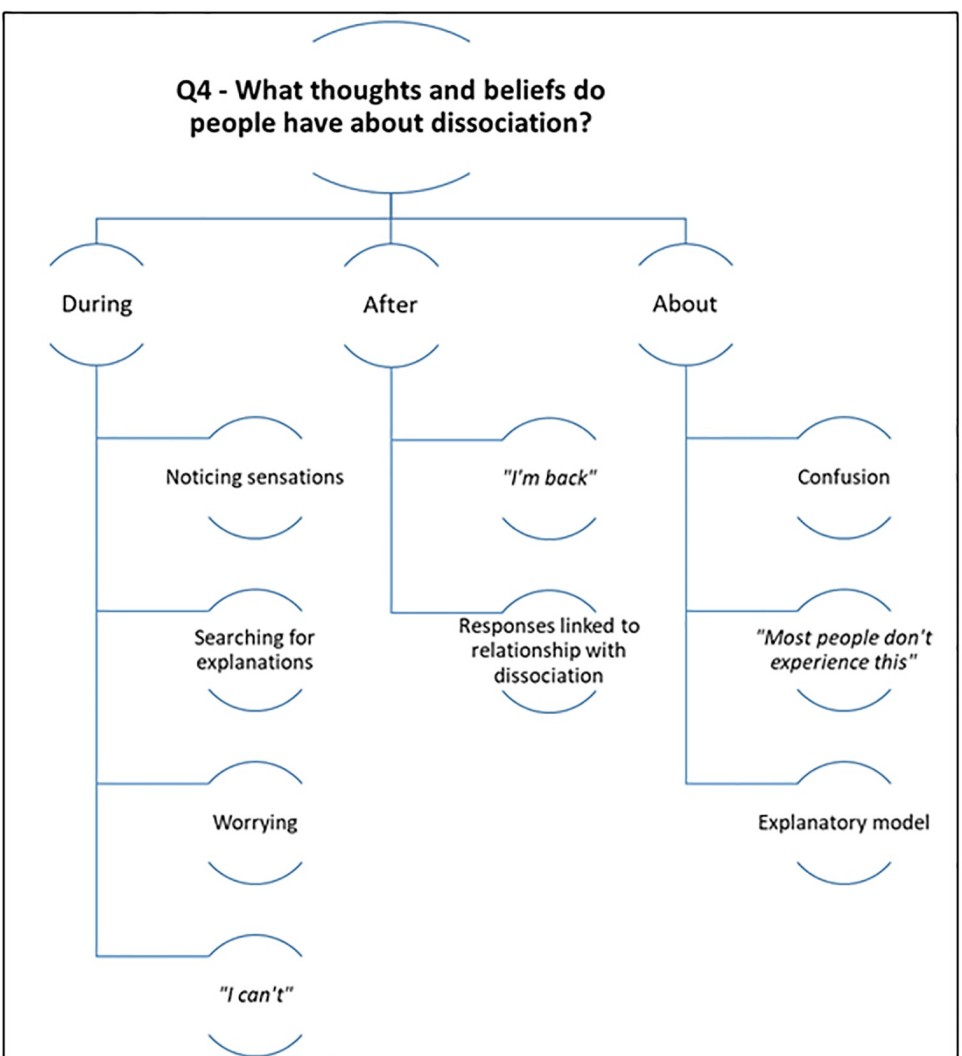

**Fig 5. Summarising the themes for the fourth research question.**

Anxious cognitions in general were frequently reported, including: worrying that dissociation was about to happen, that it meant something was wrong, worrying about the consequences of the dissociative feelings, worrying others would notice, and worrying how long the sensations would last. Thoughts about wanting the experiences to stop or end once it had begun were also reported.

Interviewees reported thoughts about being unable–or prevented from–doing the things they would usually do and not being able to cope with the sensation continuing. These thoughts appeared connected to others reflecting hopelessness, loss of motivation, or passive acceptance the situation:

P6: *"What's the point?"*

**After the experience.** Participants reported surprise (P4: *"Ooh, I'm back now"*) and relief when realising dissociation had ended. One participant reported feeling slightly *"guilty"* afterwards, linked to appraisals that it had not been "them" in control during dissociation and

therefore that they could not claim to have personally achieved what had been done during that episode. This caused internal conflict, as the participant reported knowing logically that it had been them.

"Returning" from dissociation, for some, involved checking what the impact of the episode was (P8: *"Ok, what have I missed?")*, and re-adjusting to the world after being numb or detached:

> P8: *"It's like all of a sudden, that little box that was around you isn't there now and everything's like "bam!" in your face, or in your ears, or the smell or- everything's more heightened. I struggle with that; I don't like it."*

**About the experience.**    The most common theme when asking about dissociation was that of confusion: not knowing what it is or why it happens. For some, this was acutely distressing:

> P6: *"I used to be so desperate to find out what was going on and why I was the way I was. I used to look around, look at people, look at things, as if I'm crying out 'is there somebody out there than understands what's going on to me?!'"*

The distress and confusion might be partially explained by beliefs that dissociation is unusual, or that one's mind is unusual compared to others:

> P1: *"most people don't experience this"*

Perhaps to help themselves understand what–and why–they feel this way, participants often developed *"a theory"* (P12) about dissociation. For example, Participant 5 suggests dissociation might be *"a kind of thought disorder type thing, maybe"*, and Participant 8 explains:

> P8: *"I sometimes wonder whether it's like my brain has reached the max it can cope with, so it's just like 'I'm going to shut off'"*.

Whilst these are somewhat psychological explanations, two interviewees used physical explanations: wondering if they have a brain tumour or blood clot in the brain. One participant even underwent scans to determine whether this was true. Other explanations showed the influence of hallucinations, paranoia, aberrant salience, and jumping to conclusions and appeared more psychotic-like in content–such as being trapped in a video game or computer simulation.

Explanations influenced how dangerous participants perceived dissociation to be. They also varied in conviction over time: sometimes taking on a delusional intensity, and other times acting as placeholders for experiences the participant was struggling to describe to others:

> P6: *"My description of it was as if I was the second coming of Christ. I wasn't a religious person, but that was the only way I could think of explaining it"*

## Discussion

### Key findings

The aim of the current study was to better understand dissociation in the context of non-affective psychosis. Regarding the lived experience, this appears best described as a negative, highly

subjective experience where the core feature is a "felt sense" of strangeness. The most reported 'types' of 'strangeness' in this sample were finding that things somehow felt 'unreal' or feelings of unanticipated detachment. Feeling unanticipated emotional detachment from others was associated with significant distress. People also experienced unanticipated unfamiliarity, and a sense of automaticity (rather than voluntary or effortful control).

Dissociation is conceptualised as a multi-dimensional construct, with a vast range of symptoms and experiences proposed to fall within its bounds [e.g. 19]. It is unsurprising, therefore, that the open and exploratory approach taken by this study resulted in omissions of particular experiences described in other reports of dissociation, instead bringing to the fore those symptoms that were the most problematic or most noteworthy to the people experiencing them. For example, no participant described dissociative amnesia; however, several described memory blanking and experiencing their memories as unfamiliar. In line with Lynn et al. [1], who state that 'it can be challenging, if not impossible, to distinguish amnesia from encoding failure', a number of participants stated that they were aware that their lack of memory for events related to periods when they were feeling unreal or detached. The results therefore highlight interviewees' subjective experience of their memory as somehow altered or anomalous.

It is striking that feeling unable to describe dissociative experiences was such a strong theme in the results. Participants struggled to find the words to describe dissociation, often relying on "as if" statements or metaphors to describe highly subjective and sometimes subtle experiences. The result of this difficulty–perhaps exacerbated by fear, mental avoidance, and concerns about others' reactions–was that people did not feel able to talk about these events. Lack of reporting to clinicians may partially explain why dissociation is so often under-recognised [2]. We therefore recommend that clinicians proactively ask about dissociative experiences, as clients are unlikely to describe these unprompted. Further, it was a common experience that attempting to describe dissociation resulted in the listener misunderstanding. Within this small sample of lived experience alone, clinicians reportedly interpreted descriptions of dissociation as depression, worsening psychosis, or bipolar disorder. It seems likely that people with non-affective psychosis describing dissociation to their care team repeatedly feel unheard or misunderstood, and may receive inappropriate treatment.

However, recognising the distinction between dissociation and psychosis may be especially challenging when assessing presentations featuring negative psychotic symptoms. In particular, the Positive and Negative Syndrome Scale (PANSS; [20]) negative scale items of "blunted affect", "emotional withdrawal", and "passive/apathetic social withdrawal" may present similarly to one of the main types of experience described by participants (emotional detachment) and one of their main responses (withdrawal). Further questions may therefore need to be asked when assessing a patient presenting with these symptoms in order to understand the possible cause.

## Placing the findings in context

According to interviewees, there are many interactions between dissociation and psychosis. Hallucinations or hallucination-like experiences contributed to feelings of strangeness; paranoia was identified as a potential consequence of dissociation; and a range of psychotic features (including voices) acted as triggers and modifiers for dissociation. Some readers may argue that it is difficult–or impossible–to make distinctions between psychotic and dissociative symptoms, taking the stance that these experiences overlap, and that many apparently psychotic symptoms may in fact be dissociative (and vice versa). For example, auditory hallucinations have been described as dissociative by Moskowitz and colleagues [21]. Whilst it is not within the scope of this paper to answer the question of whether or which symptoms require

re-categorising, it is important to note that participants commented on the distinguishable nature of these experiences. That is to say, people with psychosis diagnoses experience dissociation as distinct from their psychotic symptoms, describing these as having the anomalous (detached, strange, and unreal) qualities described above. Further, they reported a sense of frustration when clinicians did not understand that they were describing two separable phenomena (dissociation and psychosis).

Whilst the exact pattern of causal relationships between dissociative and psychotic experiences is unknown (whatever the extent of their overlap), these results appear to support the Garety et al [9] model of the development of positive psychotic symptoms. Namely, participants described how they found dissociative experiences confusing and distressing, and subsequently engaged in a "search for meaning". For some, the result was an explanation with a psychotic-like quality (e.g. the world is a computer simulation). Completing the chain of events proposed in the model, sometimes these explanations were held with such conviction that they met criteria for a paranoid or grandiose delusion.

Our findings also support Hunter et al.'s [22] cognitive-behavioural conceptualisation of depersonalisation and derealisation (DP/DR; a form of dissociation) within the novel context of non-affective psychosis. Hunter et al. suggest that DP/DR can result from anxiety, stress, or fatigue and is maintained by avoidance, safety behaviours, and cognitive biases (such as symptom monitoring). Our results imply additional factors may also be important: beliefs about one's ability to cope with stress or (extreme) emotion, perceived powerlessness, and ambivalence about dissociation. The results of this study may therefore be of use in the search for 'transtheoretical variables' to explain dissociation [1].

## Strengths & limitations

The results of this study are limited by its methodology: it is unclear how these findings generalise outside of the context of non-affective psychosis, for example, in other diagnoses or even in non-clinical ('non-pathological') dissociative experiences [23]. Further, participants were not asked to review transcripts for accuracy, and only a single coder was used during analysis. This reduces the validity of the results through the introduction of risk of bias. However, the sample size was adequate for a qualitative method, negative case analysis was included, and the topic guide was developed in collaboration with a Lived Experience Advisory Panel, piloted, and reviewed after the first three interviews. These elements of the methodology improve the credibility of the findings, which have good face validity and appear to triangulate with existing literature outside of psychosis [e.g. 22].

It is interesting that the DES-II scores of nine participants were in the range expected for people with dissociative disorder diagnoses, and it is a limitation of the study that co-morbid dissociative disorders were not asked about or recorded. However, given the under-recognition of such difficulties [2], it is unlikely that many of the participants had a formal diagnosis of dissociation, even if they met criteria for these. Some may argue that participants may have been mis-diagnosed with a psychotic disorder, and in fact better fit the criteria for a dissociative disorder. We believe that this is unlikely to be true for all participants, many of whom described key psychotic symptoms, such as delusions. Again, it is not the aim of this study to distinguish between psychotic or dissociative presentations other than to highlight patients' lived experience of these being separable and important in their own right. Any misdiagnosis must surely serve to further illustrate the need for clinicians to be better informed and feel more confident in recognising and asking about dissociative phenomena.

## Future directions

Based on these results, we suggest that the core subjective experience of dissociation is a 'felt sense of anomaly' (FSA). This may relate to the internal world (thoughts, emotions, or memories), the physical body, or the external world (the environment and other people). The anomaly may be a sense of unfamiliarity, unreal-ness, detachment or some other sense of deviation from normal experience. Clinicians may wish to listen out for descriptions of FSA when assessing patients, as these kinds of experiences may be suggestive of dissociation and require further exploration. Holding this common denominator in mind may therefore aid clinicians in their recognition of dissociation by simplifying a confusing array of symptoms [24] into a broad, but nevertheless descriptive heuristic.

The potential causal and maintenance factors highlighted in these results indicate specific interventions which may be helpful for ameliorating dissociation and the distress it causes. For example, stress, fatigue, avoidance, and withdrawal imply that techniques such as grounding, problem-solving, pacing, activity scheduling, and behavioural experiments may be effective in the treatment of dissociation. Motivational interviewing may be appropriate where people have high levels of ambivalence towards dissociation. Whilst some of these techniques have already been suggested as interventions for dissociation [e.g. 25], others constitute potentially novel applications and warrant further exploration.

Enhancement of interventions for dissociation may be particularly important in non-affective psychosis, given the above discussion of Garety et al. [9]. The results of this study suggest that that having a better understanding of dissociation modifies or negates the need for a "search for meaning": interviewees described seeking an explanation when they did not understand what they were experiencing, and conversely, feeling less afraid and more capable once they understood dissociation better. It is possible that psychoeducation or reducing dissociation would similarly influence the "search for meaning" implicated in the Garety et al. model. It would then be of interest to determine whether this would in turn prevent or modify the subsequent development and maintenance of psychotic experiences.

## Supporting information

**S1 File. Topic guide.**
(DOCX)

**S2 File. Reflexive statement (EČ).**
(DOCX)

## Acknowledgments

The authors would like to thank the twelve participants interviewed in this study for sharing their experiences and expertise. We also thank the McPin Foundation Lived Experience Advisory Panel for their helpful suggestions during the planning of the study, LB for his support with the pilot interview, and Ann Banks for support with transcription.

## Author Contributions

**Conceptualization:** Emma Černis, Daniel Freeman, Anke Ehlers.

**Data curation:** Emma Černis.

**Formal analysis:** Emma Černis.

**Funding acquisition:** Emma Černis.

**Investigation:** Emma Černis.

**Methodology:** Emma Černis.

**Project administration:** Emma Černis.

**Supervision:** Daniel Freeman, Anke Ehlers.

**Validation:** Daniel Freeman, Anke Ehlers.

**Writing – original draft:** Emma Černis.

**Writing – review & editing:** Emma Černis, Daniel Freeman, Anke Ehlers.

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
