## [Decision Letter · Decision Letter 0]

30 Sep 2019

PONE-D-19-23082

Describing the indescribable: A qualitative study of dissociative experiences in psychosis

PLOS ONE

Dear Dr Černis,

Thank you for submitting your manuscript to PLOS ONE. Three reviewers have read and commented on your manuscript. All of them highlighted its merit. Several problems have been highlighted not only once, and after my own reading, I agree with the reviewers' judgements. In particular, the methodology is not yet sufficiently described. It would be very important that the content analysis is described in such a way to allow others to do the same you did. At this point, I would also like to highlight that only one person coded the data, which is problematic. It does not allow the assessment of an inter-rater reliability.

Please also note that your data would need to be made accessible. 

In sum, we feel that your manuscript has merit but does not fully meet PLOS ONE’s publication criteria as it currently stands. Therefore, we invite you to submit a revised version of the manuscript that addresses the points raised during the review process.

We would appreciate receiving your revised manuscript by Nov 14 2019 11:59PM. To enhance the reproducibility of your results, we recommend that if applicable you deposit your laboratory protocols in protocols.io, where a protocol can be assigned its own identifier (DOI) such that it can be cited independently in the future. For instructions see: http://journals.plos.org/plosone/s/submission-guidelines#loc-laboratory-protocols

We look forward to receiving your revised manuscript.

Kind regards,

Christine Mohr, PhD

Academic Editor

PLOS ONE

Journal Requirements:

1. You indicated that you had ethical approval for your study. In your Methods section, please ensure you have also stated whether you obtained consent from parents or guardians of the minors included in the study or whether the research ethics committee or IRB specifically waived the need for their consent.

2. Please describe in your methods section how capacity to consent was determined for the participants in this study.

3.

We note that you have indicated that data from this study are available upon request. PLOS only allows data to be available upon request if there are legal or ethical restrictions on sharing data publicly. For information on unacceptable data access restrictions, please see http://journals.plos.org/plosone/s/data-availability#loc-unacceptable-data-access-restrictions.

Reviewers' comments:

Reviewer's Responses to Questions

**Comments to the Author**

1. Is the manuscript technically sound, and do the data support the conclusions?

Reviewer #1: Yes

Reviewer #2: Yes

Reviewer #3: Partly

2. Has the statistical analysis been performed appropriately and rigorously? 

Reviewer #1: N/A

Reviewer #2: Yes

Reviewer #3: N/A

3. Have the authors made all data underlying the findings in their manuscript fully available?

Reviewer #1: No

Reviewer #2: No

Reviewer #3: Yes

4. Is the manuscript presented in an intelligible fashion and written in standard English?

Reviewer #1: Yes

Reviewer #2: Yes

Reviewer #3: Yes

5. Review Comments to the Author

Reviewer #1: In this study the authors conducted interviews with 12 people with non-affective psychosis diagnoses regarding their experiences of dissociation. Thematic analysis was used to analyse the interview transcripts. The results cover the phenomenology of dissociation, factors that modulate dissociative experiences, and the impact of these experiences in this population.

I thoroughly enjoyed reading the manuscript, which is engaging and very well written. A qualitative approach was well suited to the aims of the study. The results are of importance for clinicians who work with people experiencing psychosis and for researchers seeking to understand dissociative experiences in psychosis, how they interact in psychotic experiences, and treatments that might ameliorate their impact. This is important because definitions of dissociation are wide ranging and it can be under-recognised in clinical practice. One weakness of the study is the lack of credibility checks of the qualitative analysis – there was only one rater and it is not clear whether team discussions were held regarding the themes or any process of checking within the team. There was also not a process of respondent checks of the themes. However, the authors do recognise this, and there are other strengths, such as lived experience input to the interview development. I believe the novelty and applicability of the results still make this an important contribution to the literature.

I just have a few very minor comments that the authors might address in order to improve the manuscript.

1. The authors list participant’s diagnoses, but is it not clear how these were ascertained – was this self-report, or based on case notes?

2. In the flow chart showing recruitment of the final sample 2 people are listed as excluded with the reason “would struggle to participate in a focused interview”. This was not in the exclusion criteria, therefore may need more explanation. Is potentially important because it means that certain people’s voices are not heard, and this was a decision made (presumably) by the researchers. Understanding this decision-making process may help to situate who the sample does (and does not) include as a result of this decision.

3. I think it would be useful to see the interview guide as supplementary material

4. It would be helpful to have some indication of the length of the interviews (average and range)

5. Given that only one researcher conducted the analysis, it might help to include a brief ‘reflexive statement’ regarding the researcher world view etc. and acknowledge how this might have influenced the construction of meanings in the analysis (e.g. does the researcher work predominantly within a CBT model? (which might have influenced constructions of meaning).

6. Line 205: I had to look up what ‘haptic perception’ was! Perhaps including a definition might be useful.

Reviewer #2: This is a very important qualitative investigation of the experience of dissociation in a group of 12 people with non-affective psychosis. I have the following comments for the authors:

It would be helpful to include data on percentages of people with dissociative disorders or symptoms in groups with psychosis rather than just “elevated” to give the reader a sense of the context of the problem.

The idea that there is a complex relationship between dissociative and psychotic symptoms is referred to throughout the whole paper. I think you need to be clearer about this and to outline, as you see it, the possible ways that dissociative and psychotic symptoms may interact. You seem to be saying that they may be different constructs that are sometimes misdiagnosed (in either direction) or that they are not necessarily different constructs (of course, there are then myriad ways in which they could be ‘overlapping”). I understand that you don’t really want to get deeply into the complexities of this in your article but the reality is you are making conclusions about it, suggesting that you do need to be clearer about all the potential ways there could be a relationship.

e.g. “Some readers may argue that it is difficult – or impossible – to make distinctions between psychotic and dissociative symptoms since they overlap, and because many apparently psychotic symptoms may in fact be dissociative (and vice versa).” This sentence suggests that you need to have covered this argument in your introduction.

A lot more detail about the method is needed. In particular the development of the topic guide and the thematic analysis method used. For example - how did the team work together to ensure the robustness of the analysis? How was the negative case analysis undertaken? More information on the process of developing the topic guide with the advisory group is needed. What lived experience did the advisory group have? How negative case analysis was undertaken should be described.

The analysis sections should be referenced more thoroughly.

It is somewhat unusual to deductively analyse by research question in a thematic analysis. In doing this then you must give a rationale for why the particular research questions chosen are central to your overall aim of furthering understanding of dissociation in those with psychosis. Why these particular four research questions should be outlined in the introduction.

More detail and rationale of the sample selection is needed:

The authors have described in detail how a decision was made regarding whether a participant met the inclusion criteria of having dissociation after having been referred to the study. An equally detailed description should be given of how participants were referred to the study. From the flow chart it seems that only three participants did not have dissociative experiences that met criteria for the study, suggesting that a lot of selection was done by non-researchers in the lead up to recruitment. This is not a criticism of the study it just needs to be clearly described as it impacts the make-up of the sample.

Three participants had symptoms on the DES under 10 and within the normal range while the rest of the sample scored over 30 (clinically significant). If a DES cut-off was to be used, why was it at the average in the normal population and not at a higher clinical cut-off? Did those participants with DES scores in the normal range appear in any way different than those in the clinical range? The bi-modal distribution within the sample needs to be discussed.

It would be helpful to describe the Garety model in more detail either in the introduction or discussion. Many readers may not be familiar with it.

The central idea derived from the thematic analysis of ‘a felt sense of anomaly” is interesting and important. I have some comments on the way you have discussed this theme: it important to be clear this this is how you have constructed dissociation to be felt/sensed etc in those with psychosis, not more generally. This is important because “anomaly’ is so central to symptom constructs in psychosis (i.e. Garety model). It’s also important to remember that some of your group (the 3 with low DES scores) may not have met any kind formal criteria for dissociative symptoms- how can we be sure that this sense of anomaly is not about psychotic symptoms?

Reviewer #3: It is an interesting and important paper but it needs revion an d more elaboration.

The method need to be more and better explaind. Which method. how- acoording to whom needs to be there.

Teh resluts section needs to be more elabortaded -the htemes should be clearly shown - preferably in a figure as suggested by Braum and Clarke which is method according to who´s method I think the authors have used.

The themes more elaborated and analysed- for example feeling unreal- a theme which is connected with disconnetedness and a theme that aslo seem to be long to when dissociation starts and when it can stop. So more working with the themes. Then I think this paper can be vey good.

6. PLOS authors have the option to publish the peer review history of their article (what does this mean?). If published, this will include your full peer review and any attached files.

Reviewer #1: No

Reviewer #2: No

Reviewer #3: Yes: Doris Nilsson

---

## [Author Response · Author response to Decision Letter 0]

20 Nov 2019

Academic Editor:

1. You indicated that you had ethical approval for your study. In your Methods section, please ensure

you have also stated whether you obtained consent from parents or guardians of the minors included in

the study or whether the research ethics committee or IRB specifically waived the need for their

consent.

Thank you for raising this important point. However, according to the UK’s National Health Service

Health Research Authority, young people aged 16 and over with capacity to consent are able to provide

informed consent for themselves: https://www.hra.nhs.uk/planning-and-improving-research/policiesstandards-

legislation/research-involving-children/ Therefore, no parental consent or waivers were

required for the 16 year old who participated in the study.

2. Please describe in your methods section how capacity to consent was determined for the participants

in this study.

This has now been clarified in the first paragraph of the Methods section.

3. We note that you have indicated that data from this study are available upon request. PLOS only

allows data to be available upon request if there are legal or ethical restrictions on sharing data

publicly. For information on unacceptable data access restrictions, please see

http://journals.plos.org/plosone/s/data-availability#loc-unacceptable-data-access-restrictions.

a) If there are ethical or legal restrictions on sharing a de-identified data set, please explain them in

detail (e.g., data contain potentially identifying or sensitive patient information) and who has imposed

them (e.g., an ethics committee). Please also provide contact information for a data access committee,

ethics committee, or other institutional body to which data requests may be sent.

b) If there are no restrictions, please upload the minimal anonymized data set necessary to replicate

your study findings as either Supporting Information files or to a stable, public repository and provide us

with the relevant URLs, DOIs, or accession numbers. Please see

http://www.bmj.com/content/340/bmj.c181.long for guidelines on how to de-identify and prepare

clinical data for publication. For a list of acceptable repositories, please see

http://journals.plos.org/plosone/s/data-availability#loc-recommended-repositories.

We have indicated that the data are not to be publically shared due to ethical restrictions. The dataset

comprises twelve transcripts of in-depth interviews with participants who describe potentially

identifying, sensitive, and highly personal information. Participants consenting to be interviewed agreed

to the study in the understanding that the details of what they have said would be kept confidential,

and were not asked whether they would consent to public sharing of their transcripts. The ethics

permissions we obtained for the study from the NHS Research Ethics Committee and Health Research

Authority therefore do not cover the sharing of the data publically.

Reviewer 1

In this study the authors conducted interviews with 12 people with non-affective psychosis diagnoses

regarding their experiences of dissociation. Thematic analysis was used to analyse the interview

transcripts. The results cover the phenomenology of dissociation, factors that modulate dissociative

experiences, and the impact of these experiences in this population.

I thoroughly enjoyed reading the manuscript, which is engaging and very well written. A qualitative

approach was well suited to the aims of the study. The results are of importance for clinicians who work

with people experiencing psychosis and for researchers seeking to understand dissociative experiences

in psychosis, how they interact in psychotic experiences, and treatments that might ameliorate their

impact. This is important because definitions of dissociation are wide ranging and it can be underrecognised

in clinical practice. One weakness of the study is the lack of credibility checks of the

qualitative analysis – there was only one rater and it is not clear whether team discussions were held

regarding the themes or any process of checking within the team. There was also not a process of

respondent checks of the themes. However, the authors do recognise this, and there are other

strengths, such as lived experience input to the interview development. I believe the novelty and

applicability of the results still make this an important contribution to the literature.

We thank the reviewer for their kind comments, and are pleased that they recognise the importance of

clarifying dissociative phenomena in such a challenging field. We agree with their comments regarding

the strengths and limitations of the study.

I just have a few very minor comments that the authors might address in order to improve the

manuscript.

1. The authors list participant’s diagnoses, but is it not clear how these were ascertained – was this selfreport,

or based on case notes?

This has now been clarified within the first paragraph of the Participants sub-section.

2. In the flow chart showing recruitment of the final sample 2 people are listed as excluded with the

reason “would struggle to participate in a focused interview”. This was not in the exclusion criteria,

therefore may need more explanation. Is potentially important because it means that certain people’s

voices are not heard, and this was a decision made (presumably) by the researchers. Understanding this

decision-making process may help to situate who the sample does (and does not) include as a result of

this decision.

A brief description of these people’s presentations has been added to the exclusion criteria paragraph

in ‘Participants’. We hope this clarifies our decision-making.

3. I think it would be useful to see the interview guide as supplementary material

This is an excellent suggestion. Please find the topic guide now appended as supplementary material

after the References section and referred to under Procedure.

4. It would be helpful to have some indication of the length of the interviews (average and range)

This detail has now been added to the Procedure sub-section.

5. Given that only one researcher conducted the analysis, it might help to include a brief ‘reflexive

statement’ regarding the researcher world view etc. and acknowledge how this might have influenced

the construction of meanings in the analysis (e.g. does the researcher work predominantly within a CBT

model? (which might have influenced constructions of meaning).

We thank the reviewer for the suggestion. A brief reflexive statement has now been added to the

supplementary material, and is referred to in the Method of Analysis section.

6. Line 205: I had to look up what ‘haptic perception’ was! Perhaps including a definition might be

useful.

This term has now been changed.

Reviewer 2

This is a very important qualitative investigation of the experience of dissociation in a group of 12

people with non-affective psychosis. I have the following comments for the authors:

It would be helpful to include data on percentages of people with dissociative disorders or symptoms in

groups with psychosis rather than just “elevated” to give the reader a sense of the context of the

problem.

Thank you, this has now been added into the introduction.

The idea that there is a complex relationship between dissociative and psychotic symptoms is referred

to throughout the whole paper. I think you need to be clearer about this and to outline, as you see it,

the possible ways that dissociative and psychotic symptoms may interact. You seem to be saying that

they may be different constructs that are sometimes misdiagnosed (in either direction) or that they are

not necessarily different constructs (of course, there are then myriad ways in which they could be

‘overlapping”). I understand that you don’t really want to get deeply into the complexities of this in

your article but the reality is you are making conclusions about it, suggesting that you do need to be

clearer about all the potential ways there could be a relationship.

e.g. “Some readers may argue that it is difficult – or impossible – to make distinctions between

psychotic and dissociative symptoms since they overlap, and because many apparently psychotic

symptoms may in fact be dissociative (and vice versa).” This sentence suggests that you need to have

covered this argument in your introduction.

Apologies, we now recognise that the sentence quoted presumes that the reader already has an

understanding of the field, and therefore is not sufficiently explained for other readers. We thank the

reviewer for highlighting this, and have rephrased sentences in the introduction and discussion to make

clear that some researchers in the field view the constructs as overlapping. We have not entered into a

full discussion of this – as the reviewer alludes to, it is outside of the remit of this paper to do so – but

also because we felt on balance that this might distract from the key message of the paper, which is

people’s lived experience and the phenomenology of dissociation.

A lot more detail about the method is needed. In particular the development of the topic guide and the

thematic analysis method used. For example - how did the team work together to ensure the

robustness of the analysis? How was the negative case analysis undertaken? More information on the

process of developing the topic guide with the advisory group is needed. What lived experience did the

advisory group have? How negative case analysis was undertaken should be described. The analysis

sections should be referenced more thoroughly.

We thank the reviewer for this opportunity to expand on the methods followed in this study. We have

now added detail in the Procedure and Method of Analysis sections.

It is somewhat unusual to deductively analyse by research question in a thematic analysis. In doing this

then you must give a rationale for why the particular research questions chosen are central to your

overall aim of furthering understanding of dissociation in those with psychosis. Why these particular

four research questions should be outlined in the introduction.

We thank the reviewer for this advice, and have now added our rationale for these research questions

to the introduction section.

More detail and rationale of the sample selection is needed:

The authors have described in detail how a decision was made regarding whether a participant met the

inclusion criteria of having dissociation after having been referred to the study. An equally detailed

description should be given of how participants were referred to the study. From the flow chart it

seems that only three participants did not have dissociative experiences that met criteria for the study,

suggesting that a lot of selection was done by non-researchers in the lead up to recruitment. This is not

a criticism of the study it just needs to be clearly described as it impacts the make-up of the sample.

Further detail regarding the referral process has been added under ‘Participants’. Unfortunately, as

much of the referral decision-making happened within NHS teams, we do not have more information

than this.

Three participants had symptoms on the DES under 10 and within the normal range while the rest of

the sample scored over 30 (clinically significant). If a DES cut-off was to be used, why was it at the

average in the normal population and not at a higher clinical cut-off? Did those participants with DES

scores in the normal range appear in any way different than those in the clinical range? The bi-modal

distribution within the sample needs to be discussed.

Participants with DES scores within the normal range were entered into the study because we were

interested in exploring a wide range of perspectives: including those who had since improved and were

no longer regularly experiencing dissociation. The participants with population-level scores were able to

recall previous dissociative experiences clearly and discuss them in detail. In these cases, the DES score

reflects that the participants had not experienced dissociation within the past two weeks, not that they

had never experienced higher levels. We have now added this detail to the Participants section.

It would be helpful to describe the Garety model in more detail either in the introduction or discussion.

Many readers may not be familiar with it.

Thank you, we have now added a brief outline of the relevant parts of this model to the introduction.

The central idea derived from the thematic analysis of ‘a felt sense of anomaly” is interesting and

important. I have some comments on the way you have discussed this theme: it important to be clear

this this is how you have constructed dissociation to be felt/sensed etc in those with psychosis, not

more generally. This is important because “anomaly’ is so central to symptom constructs in psychosis

(i.e. Garety model). It’s also important to remember that some of your group (the 3 with low DES

scores) may not have met any kind formal criteria for dissociative symptoms- how can we be sure that

this sense of anomaly is not about psychotic symptoms?

We thank the reviewer for this thoughtful response. We had intended the first paragraph of ‘Placing the

Findings in Context’ to answer concerns of this nature that a reader may have. That is, participants

reported dissociation and psychosis as distinguishable experiences, and felt frustrated when clinicians

attributed their descriptions of dissociation to their psychosis. We have reviewed this paragraph and

added further detail, which we hope addresses the reviewer’s concerns more fully.

Reviewer 3

It is an interesting and important paper but it needs revion an d more elaboration.

The method need to be more and better explaind. Which method. how- acoording to whom needs to

be there.

We hope that Reviewer 3 feels that the details added in response to Reviewers 1 and 2 sufficiently

elaborate on the methods used in this study.

Teh resluts section needs to be more elabortaded -the htemes should be clearly shown - preferably in a

figure as suggested by Braum and Clarke which is method according to who´s method I think the

authors have used.

We thank the reviewer for this helpful suggestion, which we believe will make the paper far easier for

readers to follow. Figures have been added after each section. These were originally omitted for space.

The themes more elaborated and analysed- for example feeling unreal- a theme which is connected

with disconnetedness and a theme that aslo seem to be long to when dissociation starts and when it

can stop. So more working with the themes. Then I think this paper can be vey good.

We have now added to the themes identified by the reviewer. These had been omitted for brevity, and

we thank the reviewer for the opportunity to add further detail.

---

## [Decision Letter · Decision Letter 1]

2 Jan 2020

PONE-D-19-23082R1

Describing the indescribable: A qualitative study of dissociative experiences in psychosis

PLOS ONE

Dear Dr Černis,

as you will see, the three original referees agreed to again review your revision of this manuscript. All of them were pleased with the efforts and care you have taken. While one referee is happy to accept the manuscript as is, the other two referees asked for some minor attention to the manuscript (specify the use of the DES, have a final careful read of the manuscript)

Accordingly, I am happy to communicate that your manuscript has been judged scientifically suitable for publication. At this point, I cannot  formally accept this manuscript for publication, but can informally confirm that it will be accepted once the final changes have been performed and submitted. 

Please add these very minor changes as quickly as possible to the manuscript. I am than happy to accept the manuscript. 

A marked-up copy of your manuscript that highlights changes made to the original version. This file should be uploaded as separate file and labeled 'Revised Manuscript with Track Changes'.An unmarked version of your revised paper without tracked changes. This file should be uploaded as separate file and labeled 'Manuscript'.

I am looking forward to receiving your revised manuscript.

Kind regards,

Christine Mohr, PhD

Academic Editor

PLOS ONE

Reviewers' comments:

Reviewer's Responses to Questions

**Comments to the Author**

1. If the authors have adequately addressed your comments raised in a previous round of review and you feel that this manuscript is now acceptable for publication, you may indicate that here to bypass the “Comments to the Author” section, enter your conflict of interest statement in the “Confidential to Editor” section, and submit your "Accept" recommendation.

Reviewer #1: All comments have been addressed

Reviewer #2: (No Response)

Reviewer #3: All comments have been addressed

2. Is the manuscript technically sound, and do the data support the conclusions?

Reviewer #1: (No Response)

Reviewer #2: Yes

Reviewer #3: Yes

3. Has the statistical analysis been performed appropriately and rigorously? 

Reviewer #1: (No Response)

Reviewer #2: N/A

Reviewer #3: Yes

4. Have the authors made all data underlying the findings in their manuscript fully available?

Reviewer #1: (No Response)

Reviewer #2: No

Reviewer #3: Yes

5. Is the manuscript presented in an intelligible fashion and written in standard English?

Reviewer #1: (No Response)

Reviewer #2: Yes

Reviewer #3: Yes

6. Review Comments to the Author

Reviewer #1: (No Response)

Reviewer #2: Congratulations to the authors for such thorough attention to reviewers’ comments.

I only have only one small question now in response to changes made. This line in the method implies that the DES questions were asked in relation to symptoms the past 2 weeks: “the DES score reflects that the participants had not experienced dissociation within the past two weeks, not that they had never experienced higher levels.” My understanding of the DES was that it does not require scale items to be present with last 2 weeks so I’m unclear how the DES was used in the current study. Could this be clarified please?

Reviewer #3: The authors have done a good job adressing reviewer comments. I also now think that this piece of research on dissociation has possiblity to put research forward.

Just check all the sentences once more so that there will be no mistakes.

7. PLOS authors have the option to publish the peer review history of their article (what does this mean?). If published, this will include your full peer review and any attached files.

Reviewer #1: Yes: Rachel M Brand

Reviewer #2: No

Reviewer #3: No

---

## [Author Response · Author response to Decision Letter 1]

7 Jan 2020

Reviewer 1

(No response)

Reviewer 2

Congratulations to the authors for such thorough attention to reviewers’ comments.

I only have only one small question now in response to changes made. This line in the method implies that the DES questions were asked in relation to symptoms the past 2 weeks: “the DES score reflects that the participants had not experienced dissociation within the past two weeks, not that they had never experienced higher levels.” My understanding of the DES was that it does not require scale items to be present with last 2 weeks so I’m unclear how the DES was used in the current study. Could this be clarified please?

- Thank you, we have now clarified this in the Methods section.

Reviewer 3

The authors have done a good job adressing reviewer comments. I also now think that this piece of research on dissociation has possiblity to put research forward.

Just check all the sentences once more so that there will be no mistakes.

- We thank the reviewer for their kind comments. We have proof-read the manuscript in an effort to identify any typographical errors, including ensuring that the DES-II is referred to in a consistent manner.

---

## [Editor Report · Decision Letter 2]

30 Jan 2020

Describing the indescribable: A qualitative study of dissociative experiences in psychosis

PONE-D-19-23082R2

Dear Dr. Černis,

Thank you for making the last minor changes to your manuscript. We are now pleased to inform you that your manuscript has been judged scientifically suitable for publication and will be formally accepted for publication once it complies with all outstanding technical requirements.

With kind regards,

Christine Mohr, PhD

Academic Editor

PLOS ONE
---

## [Editor Report · Acceptance letter]

4 Feb 2020

PONE-D-19-23082R2 

Describing the indescribable: A qualitative study of dissociative experiences in psychosis 

Dear Dr. Černis:

I am pleased to inform you that your manuscript has been deemed suitable for publication in PLOS ONE. Congratulations! Your manuscript is now with our production department. 

With kind regards,

on behalf of

Dr. Christine Mohr 

Academic Editor

PLOS ONE